# Multi-Grained Policy Optimization for Multi-modal Reasoning: From An Uncertainty Perspective

**GRPO**

$$w_{i,t}(\theta) = \frac{\pi_\theta\left(o_{i,t}|(I,T),o_{i,<t}\right)}{\pi_{\theta_{\text{old}}}\left(o_{i,t}|(I,T),o_{i,<t}\right)}$$

**GSPO**

$$s_i(\theta) = \exp\left(\frac{1}{|o_i|}\sum_{t=1}^{|o_i|}\log\frac{\pi_\theta\left(o_{i,t}|(I,T),o_{i,<t}\right)}{\pi_{\theta_{\text{old}}}\left(o_{i,t}|(I,T),o_{i,<t}\right)}\right)$$

**MGPO** (*ours*)

$$m_{i,t}(\theta) = (1-\alpha) \times s_i(\theta) + \alpha \times w_{i,t}(\theta)$$

$$\alpha = \begin{cases} 1 - \dfrac{k}{\beta L} \ or \ \alpha & k < \beta L \\ 0 & else \end{cases}$$

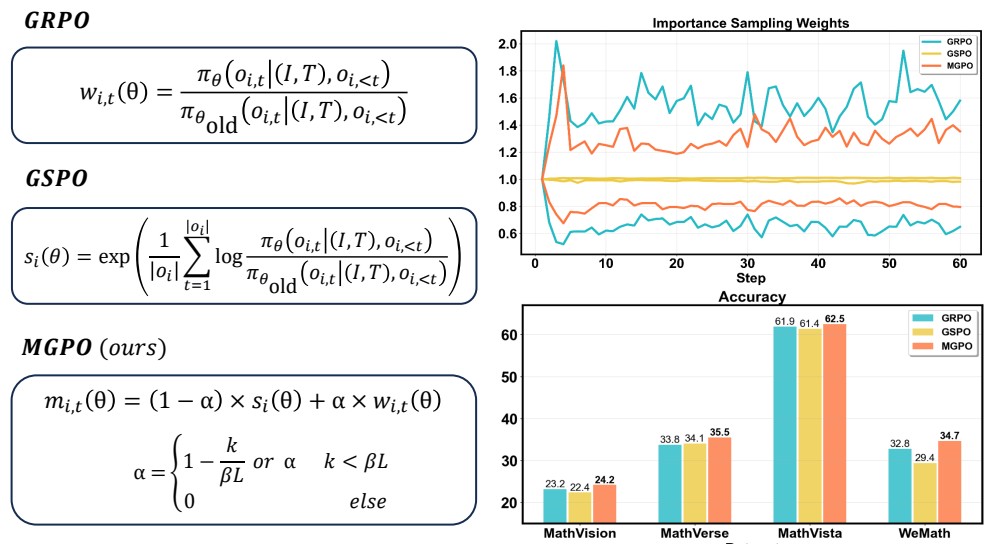

Figure 1: Comparison between importance sampling weights in Group Relative Policy Optimization (GRPO), Group Sequence Policy Optimization (GSPO) and our proposed Multi-Grained Group Policy Optimization (MGPO). GRPO utilizes token-level importance sampling weights for token exploration, while GSPO introduces sequence-level importance sampling weights with much less variation for training stabilization. MGPO derives multi-grained importance sampling weights that strike a balance between token-level and sequence-level weights based on uncertainty estimation, resulting in enhanced reasoning capabilities.

## Abstract

Reinforcement learning (RL) techniques, such as Group Relative Policy Optimization (GRPO), have substantially advanced the reasoning capabilities of Large Language Models (LLMs) and Multimodal LLMs (MLLMs). However, subsequent studies have revealed two key limitations of GRPO: training instability and insufficient token exploration in the optimization objective. To address these issues, methods like Group Sequence Policy Optimization (GSPO) introduce sequence-level importance sampling weights to mitigate training instability, and uncertainty-driven approaches emphasize low-probability tokens to encourage exploration. However, these approaches pay little attention to the balance of training stability and token exploration. In this paper, we propose *Multi-Grained Policy Optimization* (MGPO), a simple yet effective algorithm that introduces multi-grained importance sampling weights for enhanced reasoning. We first examine the effect of diverse importance sampling weights and identify their influence on training stability and token exploration during RL training. Leveraging the examination, we dynamically adjust the ratio between token-level and sequence-level importance sampling weights via uncertainty estimation on log probabilities, thereby balancing the training stability and token exploration effectively. Extensive experiments on various multimodal

reasoning benchmarks demonstrate that MGPO outperforms GRPO, GSPO, as well as multiple open-source and R1-style 3B/7B models consistently across multiple widely adopted multimodal reasoning benchmarks with **few lines of code modification**, highlighting its superior effectiveness and generalizability.

# 1 INTRODUCTION

Reinforcement Learning (RL) has recently achieved great advancements in reasoning with Large Language Models (LLMs) and Multimodal LLMs (MLLMs) (Guo et al., 2025; Team et al., 2025; Chen et al., 2025b; Huang et al., 2025). The state-of-the-art RL algorithm, such as Group Relative Policy Optimization (GRPO) (Shao et al., 2024; Guo et al., 2025) that computes the relative advantage with various generated rollouts, has demonstrated superior performance on various reasoning benchmarks. During the training process, GRPO utilizes token-level importance sampling weights to compute the relative advantage for policy updates. It adopts the clipping operation in Proximal Policy Optimization (PPO) (Schulman et al., 2017) to constrain the deviation of importance sampling weights, which stabilizes the training process.

With the rapid development of RL algorithms, two typical constraints of GRPO have been identified, namely, limited token exploration and instable training. Limited token exploration arises when low-probability tokens, despite capturing affluent information, are restricted from being sampled during policy updates. This issue has been mitigated by relaxing the clipping upper bound (Yu et al., 2025) or leveraging high-uncertainty tokens (Wang et al., 2025a). Instable training arises when the token-level importance sampling weights heavily fluctuate during policy updates, where several studies strive to stabilize the training process (Zheng et al., 2025a; Zhao et al., 2025). Group Sequence Policy Optimization (GSPO) (Zheng et al., 2025a) introduces more stable sequence-level importance sampling weights that align with sequence-level rewards. However, the more stable importance sampling weights suppress the variation, leading to degraded token exploration. This raises a critical question: **"Is it possible to harmonize training stability and token exploration within a single system?"**

To this end, this paper proposes *Multi-Grained Policy Optimization* (MGPO), a novel RL algorithm that introduces multi-grained importance sampling weights to improve training stability and token exploration simultaneously. MGPO works two novel designs. First, it derives multi-grained importance sampling weights by combining token-level and sequence-level importance sampling weights, enhancing both token exploration and training stability effectively. Second, it introduces uncertainty estimation for optimal token sampling, integrating importance sampling weights of high-uncertainty tokens. As illustrated in Figure 1, MGPO avoids both the frequent occurrence of extreme token-level importance sampling weights in GRPO, and the excessive flattening of sequence-level importance sampling weights in GSPO, leading to consistently improved reasoning capability. Extensive experiments on multiple MLLM benchmarks demonstrate that MGPO surpasses state-of-the-art models on 3B and 7B scales. Notably, MGPO consistently achieves better performance compared with vanilla GRPO and GSPO algorithms. Moreover, MGPO trained with only 2.1K samples from the open-source Geometry3K (Lu et al., 2021) dataset, surpasses early methods that require a warm-up Supervised Fine-Tuning (SFT) stage with large-scale training data (Zhang et al., 2025a; Shen et al., 2025), as well as most R1-style models (Chen et al., 2025a; Meng et al., 2025) that employ much more data during RL training. The contribution of this paper can be summarized as follows:

- We identify importance sampling weights as one key factor that could benefit both training stability and token exploration in RL training.

- A novel RL algorithm, *Multi-Grained Policy Optimization* (MGPO), is proposed to leverage uncertainty estimation to effectively integrate token-level and sequence-level importance sampling weights.

- Extensive experiments demonstrate that MGPO achieves superior performance that consistently outperforms the prevalent GRPO, GSPO, as well as most open-source and R1-style models on multimodal reasoning with few lines of code modification.

**Algorithm 1** Implementation of the Multi-Grained Importance Sampling Weights in MGPO

```
def Multi_Grained(log_probs, old_log_probs, response_mask):
    """
    log_probs [N, L]: Log probabilities of tokens from the current model
    old_log_probs [N, L]: Log probabilities of tokens from the old model
    response_mask [N, L]: Valid response tokens
    """

    # Token-level Importance Sampling Weights
    negative_approx_kl = log_probs - old_log_probs
    token_ratio = torch.exp(negative_approx_kl)

    # Sequence-level Importance Sampling Weights
    seq_lengths = torch.sum(response_mask, dim=-1)
    normalized_seq_log_prob = torch.sum(log_probs * response_mask, dim=-1) / seq_lengths
    normalized_old_seq_log_prob = torch.sum(old_log_probs * response_mask, dim=-1) /
        seq_lengths
    negative_approx_kl_seq = normalized_seq_log_prob - normalized_old_seq_log_prob
    sequence_ratio = torch.exp(negative_approx_kl_seq)

    # Multi-Grained Importance Sampling Weights
    k = int(beta * L)
    topk_values, topk_indices = torch.topk(-log_prob, k, dim=1, largest=True, sorted=True)
    thresholds = topk_values[:, -1:]
    mask = torch.where(-log_probs >= thresholds, 1.0, 0.0)

    multi_grained = sequence_ratio.unsqueeze(1).expand(N, L).clone()
    multi_grained[mask] = alpha * token_ratio[mask] + (1 - alpha) * sequence_ratio[mask]
    return multi_grained
```

## 2 RELATED WORK

### 2.1 REINFORCEMENT LEARNING IN REASONING LLMS AND MLLMS

Reinforcement learning (RL) has emerged as a promising paradigm for the post-training of Large Language Models (LLMs). RL is originally designed for alignment via Human Feedback (RLHF) (Ouyang et al., 2022; Achiam et al., 2023; Yu et al., 2024), and has recently been developed for Verifiable Reward (RLVR) (Shao et al., 2024; Guo et al., 2025; Team et al., 2025) that demonstrates great potential in enhanced reasoning capability. Originating from Proximal Policy Optimization (PPO) (Schulman et al., 2017), several algorithmic variants have been proposed, including GRPO (Shao et al., 2024), DAPO (Yu et al., 2025), Dr.GRPO (Liu et al., 2025b), EMPO (Zhang et al., 2025b), GSPO (Zheng et al., 2025a), GMPO (Zhao et al., 2025), with the aim of improving efficiency, stability and scalability.

Beyond LLMs, the reasoning capability of Multimodal Large Language Models (MLLMs) has also been widely investigated. Early attempts in multimodal domain focus on constructing R1-style RL training methods, such as R1-V (Chen et al., 2025b), LMM-R1 (Peng et al., 2025b), Vision-R1 (Huang et al., 2025), VLM-R1 (Shen et al., 2025), etc. Subsequent studies have shifted focus to RL algorithm designs, such as specific reward functions (Zhang et al., 2025a; Peng et al., 2025a), training strategies (Chen et al., 2025a; Wang et al., 2025b), and data augmentation schemes (Liu et al., 2025a; Yao et al., 2025). Despite these advancements, the balance between token exploration and training stability remains largely unexplored in the multimodal domain, which is essential for generalization on various multimodal reasoning tasks.

### 2.2 UNCERTAINTY ESTIMATION IN REASONING LLMS AND MLLMS

Recent studies have leveraged **entropy** as the uncertainty estimation measure to revitalize training. EMPO (Zhang et al., 2025b) claims that the semantic entropy is negatively correlated with model accuracy, suggesting that entropy minimization incentivizes reasoning capability. Later studies, such as SEED-GRPO (Chen et al., 2025c) compute semantic entropy among various rollouts to modulate the magnitude of policy updates. 80/20 rule (Wang et al., 2025a) highlights the crucial impact of minority tokens with high entropy for enhanced LLM reasoning capability. (Cheng et al., 2025) inserts entropy regularization into the advantage term for better exploration of longer reasoning chains. Despite these advancements, existing work neglects uncertainty on importance sampling weights, leaving an significant aspect unaddressed.

# 3 METHOD

In this section, we first introduce the preliminaries of GRPO (Shao et al., 2024) and GSPO (Zheng et al., 2025a). Then, we propose our method MGPO, encouraging dynamic integration between token-level and sequence-level importance sampling weights for enhanced MLLM reasoning. Guided by the log probabilities as token uncertainty, MGPO dynamically adjusts the importance sampling weights end-to-end during RL training with few lines of code modification. The overall algorithm is provided in Algorithm 1.

## 3.1 PRELIMINARIES

**Group Relative Policy Optimization (GRPO).** GRPO (Shao et al., 2024) is a variant of Proximal Policy Optimization (PPO) (Schulman et al., 2017) that removes the requirement of the value and reward model for efficient RL. For the multimodal setting, it first uses a pre-trained MLLM as the initial policy model $\pi_\theta$ and the reference model $\pi_{\theta_{\mathrm{ref}}}$. For a given image-text pair $(I, T)$ from the training set, the old policy model $\pi_{\theta_{\mathrm{old}}}$ generates $N$ rollouts $\{o_1, o_2, ..., o_N\}$. A verifiable reward function is designed and utilized to calculate the corresponding rewards for each rollout $\{R_1, R_2, ..., R_N\}$, and the relative advantage $\widehat{A}_{i,t}$ can be defined as follows,

$$\widehat{A}_{i,t} = \widehat{A}_i = \frac{R_i - \mathrm{mean}\left(\{R_i\}_{i=1}^N\right)}{\mathrm{std}\left(\{R_i\}_{i=1}^N\right)}, \quad w_{i,t}(\theta) = \frac{\pi_\theta(o_{i,t}|(I,T), o_{i,<t})}{\pi_{\theta_{\mathrm{old}}}(o_{i,t}|(I,T), o_{i,<t})}, \tag{1}$$

where $w_{i,t}(\theta)$ denotes the token-level importance sampling weights, which facilitate effective transformation between current and old policy models, and thus enhance sample efficiency and improve training stability. The overall objective of GRPO can be defined as,

$$\mathcal{J}_{\mathrm{GRPO}}(\theta) = \mathbb{E}_{(I,T)\sim\mathcal{D}, \{o_i\}_{i=1}^N\sim\pi_{\theta_{\mathrm{old}}}(\cdot|(I,T))}$$
$$\left[\frac{1}{N}\sum_{i=1}^N \frac{1}{|o_i|}\sum_{t=1}^{|o_i|}\min\left(w_{i,t}(\theta)\widehat{A}_{i,t}, \mathrm{clip}\left(w_{i,t}(\theta), 1-\varepsilon, 1+\varepsilon\right)\widehat{A}_{i,t}\right)\right], \tag{2}$$

where $\epsilon$ sets the clipping range. Note that the KL divergence term $\mathbb{D}_{\mathrm{KL}}[\pi_\theta|\pi_{\theta_{\mathrm{ref}}}]$ is omitted following recent papers (Meng et al., 2025; Liu et al., 2025b) for better performance.

**Group Sequence Policy Optimization (GSPO).** GSPO (Zheng et al., 2025a) reveals that the key weakness in GRPO is the mismatch between the unit of the optimization objective and the reward function. As the reward function is sequence-level computed, the token-level importance sampling weights cause high-variance noise during training. Thus, GSPO introduces the sequence-level importance sampling weights,

$$s_i(\theta) = \exp\left(\frac{1}{|o_i|}\sum_{t=1}^{|o_i|}\log\frac{\pi_\theta(o_{i,t}|(I,T), o_{i,<t})}{\pi_{\theta_{\mathrm{old}}}(o_{i,t}|(I,T), o_{i,<t})}\right). \tag{3}$$

The objective of GSPO can be defined as,

$$\mathcal{J}_{\mathrm{GSPO}}(\theta) = \mathbb{E}_{(I,T)\sim\mathcal{D}, \{o_i\}_{i=1}^N\sim\pi_{\theta_{\mathrm{old}}}(\cdot|(I,T))}\left[\frac{1}{N}\sum_{i=1}^N\min\left(s_i(\theta)\widehat{A}_i, \mathrm{clip}\left(s_i(\theta), 1-\varepsilon, 1+\varepsilon\right)\widehat{A}_i\right)\right], \tag{4}$$

## 3.2 MULTI-GRAINED POLICY OPTIMIZATION

While GSPO introduces the sequence-level importance sampling weights $s_i(\theta)$ to partially mitigate the fluctuation caused by noise in $w_{i,t}(\theta)$, we rethink whether $w_{i,t}(\theta)$ possesses information that $s_i(\theta)$ lacks. From the theoretical perspective, $s_i(\theta)$ estimates how far the **overall response** $o_i$ sampled from $\pi_{\theta_{\mathrm{old}}}$ deviates from $\pi_\theta$, while $w_{i,t}(\theta)$ is capable to estimate how far the **current token** $o_{i,t}$ sampled from $\pi_{\theta_{\mathrm{old}}}$ deviates from $\pi_\theta$. Under a more fine-grained behavior, the token-level importance sampling weights can timely impel the policy model for token-level corrections. Therefore, we believe that the complementarity of these two importance sampling weights could yield better performance.

Based on the aforementioned insights, we propose **_Multi-Grained Policy Optimization_** (MGPO), to dynamically integrate the importance sampling weights of $w_{i,t}(\theta)$ and $s_i(\theta)$.

**Multi-grained Integration.** The token-level importance sampling weights $w_{i,t}(\theta) \in \mathbb{R}^{N \times L}$ in GRPO reflects an unequal distribution of the log likelihoods for all tokens in each rollout, while the sequence-level importance sampling weights $s_i(\theta) \in \mathbb{R}^N$ in GSPO is equally distributed among all tokens in each rollout, where $L$ denotes the response length. Thus, a straightforward attempt is to directly mix the importance sampling weights at token-level,

$$m_{i,t}(\theta) = \alpha \times w_{i,t}(\theta) + (1 - \alpha) \times s_{i,t}(\theta) \tag{5}$$

where $s_{i,t}(\theta) \in \mathbb{R}^{N \times L}$ is the token-level expansion of $s_i(\theta)$, and $\alpha$ denotes the blended ratio. The blended ratio $\alpha$ can be defined either as a fixed value shared across all tokens, or as a dynamic value that adapts throughout training to enable more flexible policy updates. From this perspective, we investigate an optimal $\alpha$ to effectively integrate the two importance sampling weights according to the statistics of the current policy. Inspired by uncertainty estimation techniques in reinforcement learning (Wang et al., 2025a; Cheng et al., 2025), we posit that policy updates should place greater emphasis on high-uncertainty tokens. Consequently, we adopt token-level uncertainty as the metric to refine the $\alpha$ for integration, and the details are as follows.

**Uncertainty-aware Token Sampling.** $P_{i,t} = \pi_\theta(o_{i,t} \mid (I, T), o_{i,<t})$ denotes the logits probability, reflecting the log likelihood of generating the $t$-th token at its corresponding index. Accordingly, $-P_{i,t}$ serves as a measure of uncertainty: smaller value corresponds to the more confident tokens, while larger values indicate higher uncertainty. We therefore employ $-P_{i,t}$ as the uncertainty score to guide token sampling for improved token exploration.

For each rollout $o_i \in \mathbb{R}^L$ with corresponding token uncertainty $-P_{i,t}$, we rank all tokens by $-P_{i,t}$ from highest to lowest. Tokens with larger uncertainty values are regarded as high-uncertainty tokens, which should be retained with greater probability to encourage exploration. Consequently, for the top-$\beta$ ($\beta \in [0, 1]$) tokens, we apply Eq. 5 to integrate token-level and sequence-level importance sampling weights, thereby timely exploiting high-uncertainty tokens to facilitate effective policy updates. In contrast, the remaining tokens are assigned sequence-level importance sampling weights to stabilize training.

Based on the aforementioned paradigm, we further propose a fixed and a dynamic $\alpha$ as shown in Eq. 6. For the fixed $\alpha$, we encourage exploration on high-uncertainty tokens with an identical value for integration. For the dynamic $\alpha$, we exploit the ranked uncertainty scores as a control signal. Specifically, high-uncertainty tokens are assigned relatively larger $\alpha$ values, encouraging exploration through $w_{i,t}(\theta)$, whereas low-uncertainty tokens gradually allocate more weight to $s_{i,t}(\theta)$, thus prioritizing stability in policy updates,

$$\alpha = \begin{cases} \alpha, & k \le \beta L \\ 0, & k > \beta L \end{cases} \qquad \alpha = \begin{cases} 1 - k/(\beta L), & k \le \beta L \\ 0, & k > \beta L \end{cases} \tag{6}$$

where $k \in [0, L-1]$ denotes the rank index of the token, and a smaller $k$ indicates the index of a higher uncertainty token. Accordingly, the clipped multi-grained importance sampling weights can be defined as:

$$m_{i,t}^{clip}(\theta) = \alpha \times \text{clip}\left(s_{i,t}(\theta), 1-\varepsilon, 1+\varepsilon\right) + (1-\alpha) \times \text{clip}\left(w_{i,t}(\theta), 1-\varepsilon, 1+\varepsilon\right) \tag{7}$$

With the definition of multi-grained importance sampling weights, the MGPO objective function is defined as,

$$\mathcal{J}_{\text{MGPO}}(\theta) = \mathbb{E}_{(I,T) \sim \mathcal{D}, \{o_i\}_{i=1}^N \sim \pi_{\theta_{\text{old}}}(\cdot|(I,T))} \left[ \frac{1}{N} \sum_{i=1}^N \frac{1}{|o_i|} \sum_{t=1}^{|o_i|} \min\left( m_{i,t}(\theta) \widehat{A}_{i,t}, m_{i,t}^{clip}(\theta) \widehat{A}_{i,t}, \right) \right], \tag{8}$$

## 4 EXPERIMENTS

**Dataset.** The training dataset for all models is Geometry3K (Lu et al., 2021), with a total number of 2.1K training samples. The evaluation datasets consist of four out-of-distribution multimodal

Table 1: Comparison with prevalent 3B VLMs on five out-of-distribution multimodal reasoning datasets. "*" indicates models evaluated using the official VLMEvalKit (Duan et al., 2024) by ours, while others are the reported results in corresponding papers. Data sizes used for SFT and RL are listed as SFT+RL. The best and second best values are respectively highlighted in **bold** and underlined.

| Model | Data Size | MathVerse | MathVision | MathVista | WeMath | Average |
|---|---|---|---|---|---|---|
| *Open-Source Models* | | | | | | |
| InternVL3-2B (Zhu et al., 2025) | - | 24.5 | 20.2 | 57.6 | 22.9 | 31.3 |
| SAIL-VL-1.5-2B (Dong et al., 2025) | - | 20.9 | 17.9 | **67.0** | 16.7 | 30.6 |
| *R1-style Models* | | | | | | |
| R1-VL-2B (Zhang et al., 2025a) | 260K+10K | 26.2 | - | 52.1 | - | - |
| VLM-R1-3B (Shen et al., 2025) | - | 32.2 | 21.9 | 62.7 | 30.0 | 36.7 |
| VLAA-Thinker-3B (Chen et al., 2025a) | 25K | **36.4** | **24.4** | 61.0 | 33.8 | 38.9 |
| Qwen2.5-VL-3B-Instruct* (Bai et al., 2025) | - | 30.2 | 22.3 | 62.4 | 24.1 | 34.8 |
| + vanilla GRPO* (Shao et al., 2024) | 2.1K (Geometry3K) | 33.8 | 23.2 | 61.9 | 32.8 | 37.9 |
| + vanilla GSPO* (Zheng et al., 2025a) | 2.1K (Geometry3K) | 34.1 | 22.4 | 61.4 | 29.4 | 36.8 |
| + Ours* | 2.1K (Geometry3K) | 35.5 | 24.2 | 62.5 | **34.7** | **39.2** |

Table 2: Comparison with prevalent 7B VLMs on four out-of-distribution multimodal reasoning datasets. Other notations are consistent with Table 1.

| Model | Data Size | MathVerse | MathVision | MathVista | WeMath | Average |
|---|---|---|---|---|---|---|
| *Open-Source Models* | | | | | | |
| InternVL3-8B (Zhu et al., 2025) | - | 38.5 | **30.0** | 70.5 | 39.5 | 44.6 |
| LLaVA-OneVision-7B (Li et al., 2024) | - | 26.2 | - | 63.2 | - | - |
| Kimi-VL-16B (Kimi Team, 2025) | - | 44.9 | 21.4 | 68.7 | - | - |
| URSA-8B (Luo et al., 2025) | - | 45.7 | 26.2 | 59.8 | - | - |
| *R1-style Models* | | | | | | |
| R1-VL-7B (Zhang et al., 2025a) | 260K+10K | 40.0 | 24.7 | 63.5 | - | - |
| Vision-R1-7B (Huang et al., 2025) | 200K+10K | **52.4** | - | **73.5** | - | - |
| R1-OneVision-7B (Li et al., 2024) | 155K+10K | 47.1 | 23.5 | 64.1 | 61.8 | 49.1 |
| OpenVLThinker-7B (Deng et al., 2025) | 35K+15K | 47.9 | 25.3 | 70.2 | 64.3 | 51.9 |
| VLAA-Thinker-7B* (Chen et al., 2025a) | 25K | 50.6 | 26.4 | 68.0 | 61.5 | 51.6 |
| ADORA-7B (Gui & Ren, 2025) | 2.1K | 50.1 | 23.0 | **73.5** | 64.2 | 52.7 |
| Qwen2.5-VL-7B-Instruct* (Bai et al., 2025) | - | 42.7 | 25.7 | 68.3 | 35.9 | 43.2 |
| + vanilla GRPO* (Shao et al., 2024) | 2.1K (Geometry3K) | 50.3 | 27.6 | 69.1 | 63.1 | 52.5 |
| + vanilla GSPO* (Zheng et al., 2025a) | 2.1K (Geometry3K) | 48.6 | 25.1 | 67.3 | 61.0 | 50.5 |
| + Ours* | 2.1K (Geometry3K) | 50.9 | 27.0 | 71.2 | **65.0** | **53.5** |

reasoning benchmarks, including MathVerse (Zhang et al., 2024), MathVision (Wang et al., 2024), MathVista (Lu et al., 2023) and WeMath (Qiao et al., 2024).

**Implementation Details.** The base model for training is Qwen2.5-VL-3B/7B-Instruct (Bai et al., 2025). We use EasyR1 (Zheng et al., 2025b) as the training framework, which is built on VeRL (Sheng et al., 2024) that exclusively designed for VLMs. Most hyper-parameters remain consistent with EasyR1, including a rollout batch size of 512, rollout temperature of 1.0, learning rate of 1e-6. For training efficiency, we generate 5 rollouts for 3B model, while 10 rollouts for 7B model on all the re-implemented algorithms for fair comparison. For specific configurations, we set the blended ratio $\alpha = 0.5$ and adopt $\beta = 0.5$.

## 4.1 MAIN RESULTS

**Comparison with state-of-the-art models.** Compared with open-source baselines and recent R1-style methods, our approach achieves superior performance at 3B scale. As shown in Table 1, our method yields substantial gains on four out-of-distribution datasets, with an average improvement of 4.4% on the strong base model Qwen2.5-VL-3B-Instruct. Results on 7B models in Table 2, further confirm this advantage: our method achieves 50.9% on MathVerse, 71.2% on MathVista, surpassing most existing state-of-the-art R1-style methods, such as ADORA-7B by an average of 0.8%.

**GRPO *vs.* GSPO *vs.* Ours.** We further compare our approach against baseline RL algorithms, GRPO and GSPO at both the 3B and 7B scales. For 3B models, our method consistently outperforms GRPO and GSPO, achieving an improvement of 1.7% and 1.4% on MathVerse, and 1.9% and 5.3% on WeMath, respectively. At the 7B scale, both GRPO and GSPO have a stronger effect on vanilla Qwen2.5-VL-7B-Instruct, with GRPO in particular yielding 9.3% on average. Nevertheless, our method still improves on GRPO and GSPO by 1.0% and 3.0% on average, while incurring negligible additional training cost, thus validating the effectiveness and efficiency of our design.

**Training efficiency with limited data.** A key strength of our approach lies in its exceptional data efficiency. Using only 2.1K training samples from Geometry3K, our method achieves substantially stronger results than prior R1-style approaches. For instance, R1-VL and R1-OneVision rely on an additional SFT warm-up stage with over 200K training samples, yet attain only 40.0% and 46.1% on MathVerse. More recent approaches, such as VLAA-Thinker-7B, require over 10K training samples but still underperform our method. In contrast, by simply adjusting the importance sampling weights with few lines of code modification, our method achieves superior performance with dramatically fewer samples, highlighting both its efficiency and scalability.

## 4.2 ABLATION STUDY

**Effectiveness of different components.** Our approach consists of two key components: Multi-Grained Integration (MGI) mechanism and Uncertainty-aware Token Sampling (UTS) strategy. To evaluate their contributions, we begin by re-implementing the vanilla GRPO algorithm, which already achieves an improvement of 8.7% and 0.9% on WeMath and MathVision over the base model Qwen2.5-VL-3B-Instruct, as shown in Table 3. Building upon this, we introduce

Table 3: Ablation study on different components.

| Method | WeMath | MathVision |
|---|---|---|
| Qwen2.5-VL-7B-Instruct | 24.1 | 22.3 |
| + GRPO | 32.8 | 23.2 |
| + ours w.o. UTS | 33.6 | 23.7 |
| + MGPO | **34.7** | **24.2** |

the MGI mechanism across all tokens, which improves GRPO by 0.8% on WeMath and 0.5% on MathVision, suggesting that the multi-grained importance sampling weights benefit vanilla GRPO. With further UTS strategy, which selectively emphasizes high-uncertainty tokens based on uncertainty estimation, our method yields additional improvements of 1.1% and 0.5% on WeMath and MathVision, respectively. These results underscore the complementary roles of MGI and UTS, particularly highlighting the effectiveness of leveraging uncertainty estimation during policy updates.

**Blended Ratio $\alpha$.** We investigate the blended ratio $\alpha$ for effective integration of token-level and sequence-level importance sampling weights. As illustrated in Figure 2, the optimal fixed value of $\alpha$ is 0.5, while the performance degrades when $\alpha = 0$, which corresponds to the vanilla GSPO. Moreover, the dynamic $\alpha$ that smoothly integrates multi-grained weights in Eq. 6, achieving comparable performance of $\alpha = 0.5$, highlighting the strong generalizability of our paradigm.

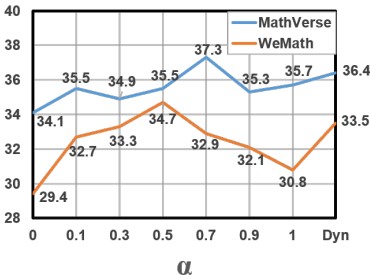

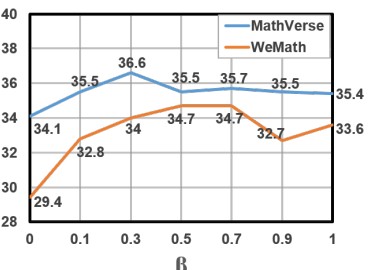

Figure 2: Experimental results on the blended ratio $\alpha$ for integration of token-level and sequence-level importance sampling weights. 'Dyn' denotes the dynamic $\alpha$ in Eq. 6.

Figure 3: Experimental results on hyper-parameter $\beta$ for token sampling ratio based on uncertainty estimation.

**Uncertainty-aware Sampling Ratio $\beta$.** Since our method samples tokens based on uncertainty estimation, we conduct an ablation study to examine the effect of the sampling ratio $\beta$. As shown in Figure 3, our approach consistently outperforms the baseline GRPO and GSPO methods across different values of $\beta$, demonstrating the robustness of $\beta$. The optimal performance is achieved when $\beta = 0.5$, suggesting that sampling a certain fraction of high-uncertainty tokens strikes a desirable balance: it avoids the excessive flattening of importance sampling weights as observed in GSPO, while mitigating the instability issues inherent to GRPO.

**The effect of temperature $\tau$ and clipping ratio $\varepsilon_{\text{high}}$.** We show the effect of temperature $\tau$ for decoding in Figure 4. From the theoretical perspective, a larger temperature indicates a more smooth logits distribution, which generates a diversified rollouts, and vice versa. The results indicate that excessively high or low values of $\tau$ negatively impact model performance, with the optimal setting achieved at $\tau = 1$ by default. Meanwhile, Table 4 shows that the clipping ratio in MGPO attains optimal performance at a moderate value $\varepsilon_{\text{high}} = 0.2$. This suggests that excessively large $\varepsilon_{\text{high}}$, such as $\varepsilon_{\text{high}} = 0.28$ in DAPO, may allow extreme multi-grained importance sampling weights and compromise stability for MGPO, whereas small $\varepsilon_{\text{high}}$ restricts token exploration.

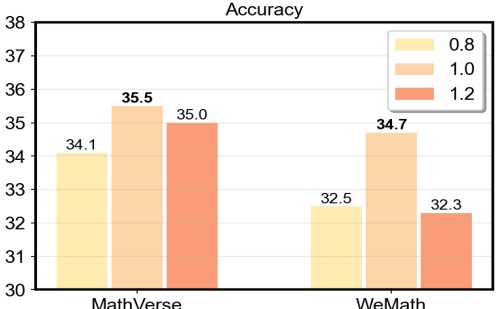

Figure 4: Experimental results of temperature $\tau$ for decoding during training.

Table 4: Experimental results on the upper bound of the clipping ratio $\varepsilon_{\text{high}}$ during training.

| Method | MathVerse | WeMath |
|---|---|---|
| Qwen2.5-VL-3B-Instruct | 30.2 | 24.1 |
| + MGPO ($\varepsilon_{\text{high}} = 0.15$) | 35.9 | 31.3 |
| + MGPO ($\varepsilon_{\text{high}} = 0.2$) | **35.7** | **34.7** |
| + MGPO ($\varepsilon_{\text{high}} = 0.25$) | 35.7 | 33.1 |
| + MGPO ($\varepsilon_{\text{high}} = 0.28$) | 34.0 | 33.1 |

### 4.3 DETAILED ANALYSIS

**Importance Sampling Weights.** The key of our design lies in the multi-grained importance sampling weights that integrate token-level and sequence-level counterparts. For better illustration, we visualize the gap between the maximum and minimum values of the importance sampling weights for GRPO, GSPO, and our method, as shown in Figure 5. The results reveal that GRPO exhibits sharp spikes during training, where extreme values may compromise training stability. In contrast, GSPO produces overly flattened curves, which diminish the contribution of high-uncertainty tokens and risk information loss. Our method effectively balances these two extremes by retaining informative token-level weights for high-uncertainty tokens while simultaneously ensuring stability through sequence-level weights.

**Response Length.** We further compare the response length produced by GRPO, GSPO, and our method. As shown in Figure 5, our approach achieves the shortest average response length, with an average of 241 tokens per response. Particularly, GSPO generates the longest responses. This can be attributed to the under-emphasis of high-uncertainty tokens during training, thereby weakening the optimization of important tokens that leads to verbose outputs. In contrast, the curves of GRPO and our method exhibit similar trends, suggesting that our approach preserves the information of token-level importance sampling weights by enhancing the emphasis on high-uncertainty tokens. As a result, MGPO presents a more concise and informative response to avoid overthinking in multimodal reasoning.

**Entropy Loss.** Figure 5 illustrates the entropy loss trajectories of GRPO, GSPO, and our method. Our approach yields an intermediate entropy level, striking a balance between the two baselines. Compared with GRPO, it exhibits a steeper decline, indicating faster and more stable convergence; Compared with GSPO, it maintains a smoother trend that promotes token exploration. These results underscore the strength of our integration design, which effectively balances stability and exploration by reducing uncertainty while preserving sufficient diversity during training.

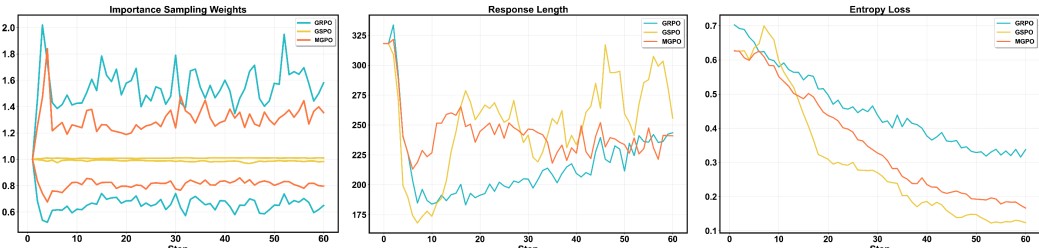

Figure 5: Comparison of GRPO, GSPO and our method in terms of training dynamics, including the importance sampling weights, response length and entropy loss.

**Case Study.** We present an illustrative example of mathematical reasoning questions in Figure 6. Compared with GRPO and GSPO, our approach is able to correctly parse the problem statements and generate coherent reasoning traces that lead to a correct answer. In contrast, GRPO and GSPO often produce convoluted reasoning paths, accompanied by incorrect solutions and over-length responses. For example, GRPO misinterprets the notion of 'base area' in the formula for the volume of a cone, leading to an incorrect answer and ultimately output repetition. Similarly, GSPO confuses the definitions of 'radius' and 'diameter', resulting in an incorrect answer. In contrast, our method accurately analyzes each reasoning trace and produces the correct answer. The case further demonstrates that our method achieves greater reliability in reasoning. Refer to Appendix A.1 for more illustrative cases.

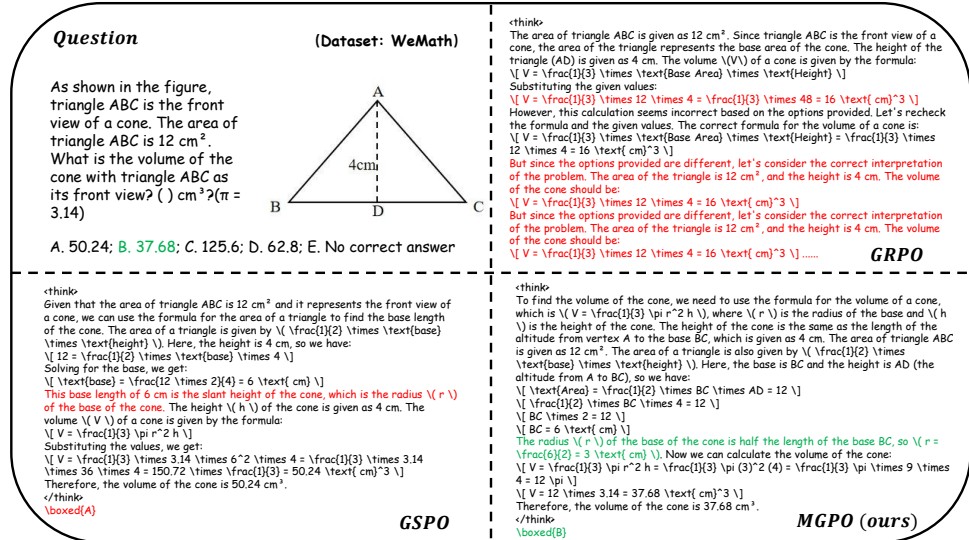

Figure 6: Case visualization of our method over vanilla GRPO and GSPO on reasoning capability.

## 5 CONCLUSION

In this paper, we investigate the balance between token exploration and training stability, and propose *Multi-Grained Policy Optimization* (MGPO), a simple yet effective algorithm that introduces multi-grained importance sampling weights from both token-level and sequence-level with the guidance of uncertainty estimation. Extensive experiments demonstrate that MGPO consistently outperforms strong baselines GRPO and GSPO, as well as open-source state-of-the-art models for multimodal reasoning. Comprehensive analyses further reveal its advantages in terms of training efficiency, response length, and entropy reduction, providing additional evidence of the robustness and effectiveness of our approach.

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

# A  APPENDIX

## A.1  ADDITIONAL CASE STUDY

We visualize additional cases to further demonstrate the advantages of our proposed MGPO over GRPO and GSPO, while also highlighting several failure cases to guide future research.

As illustrated in Figure 7, the question involves intentionally confusing conditions, such as the connector 'AB', which misleads both GRPO and GSPO into an incorrect reasoning path. In contrast, our method successfully identifies the key concept of an 'isosceles trapezoid' and produces the correct answer with a shorter token length, reflecting higher reasoning efficiency. However, when confronted with more complex problems, such as Figure 8, our method still struggles in a manner similar to that of GRPO and GSPO, drifting into an incorrect reasoning direction. This limitation suggests the potential of incorporating additional high-quality CoT datasets for supervised fine-tuning.

The MathVerse Vision Only dataset poses an additional challenge, since each problem is presented solely as an image that contains both the question and the diagram. Nevertheless, our method exhibits strong reasoning capability by directly extracting the key information for the solution. For instance, as shown in Figure 9, MGPO correctly interprets the given conditions to compute 'angle C' as the answer, while GRPO and GSPO hallucinate a false equivalence of 'angle B = angle CDE', leading to an incorrect answer. Similarly in Figure 10, our method successfully leverages the geometric property that the angle at the center of an arc is twice the angle at the circumference, whereas GRPO and GSPO misinterpret this relation and fail to solve the problem. However, during the reasoning process,

our method occasionally introduces minor errors in the reasoning path (e.g., mislabeling angle OBA instead of angle AOB). Furthermore, in Figure 11, all three methods misidentify a critical angle in the diagram, which may be due to the limited performance of the vision encoder.

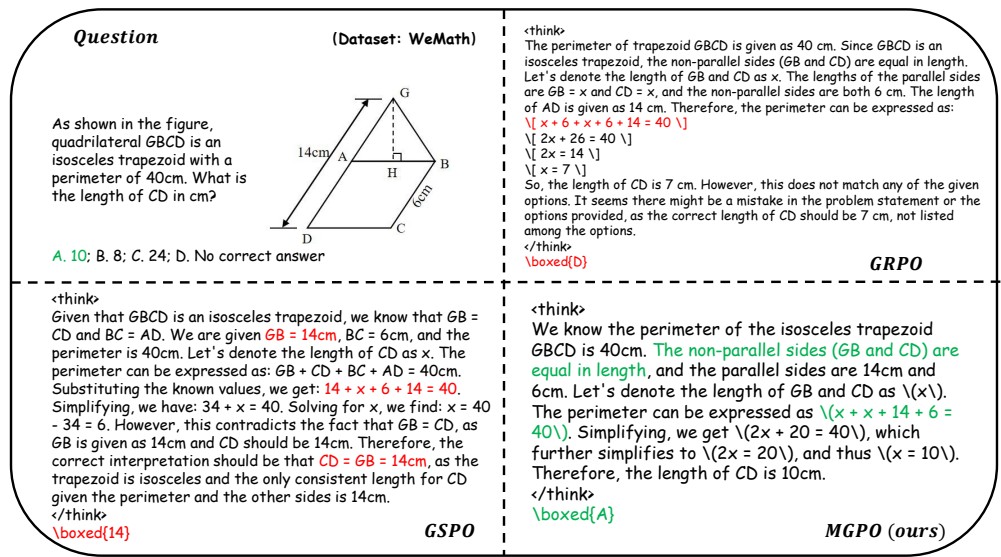

Figure 7: Case visualization of our method over vanilla GRPO and GSPO on the WeMath benchmark.

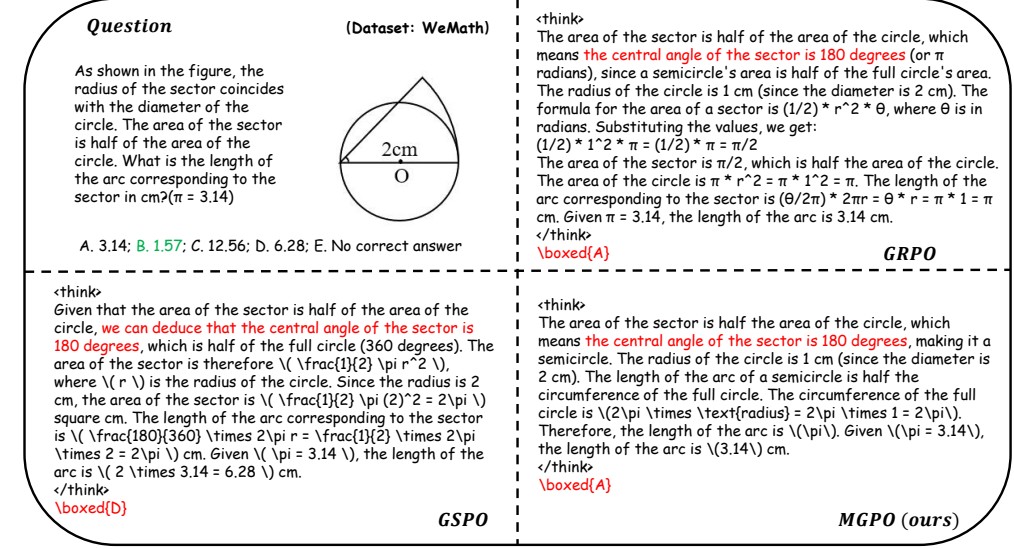

Figure 8: Case visualization of our method over vanilla GRPO and GSPO on the WeMath benchmark.

## A.2 LLMs Usage

We only use LLMs to polish writing of our manuscript. There's no other usage of LLMs in the paper, such as retrieval and discovery, and research ideation.

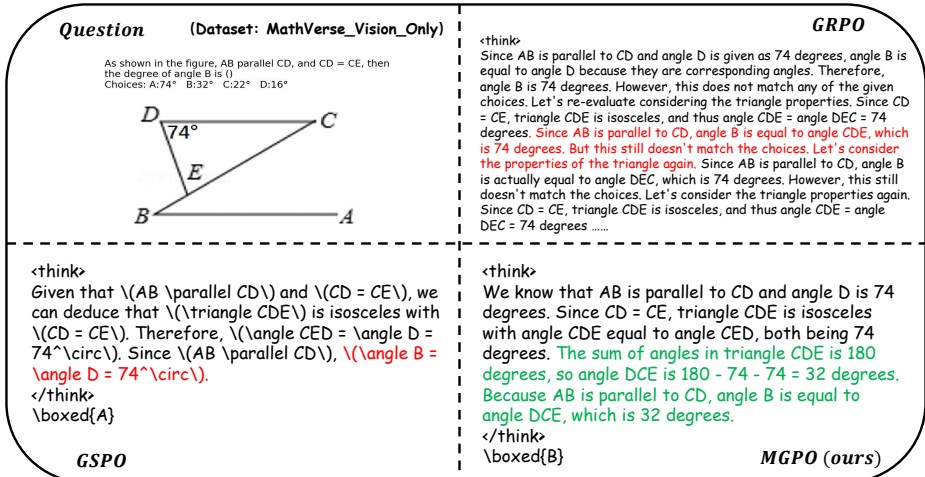

Figure 9: Case visualization of our method over vanilla GRPO and GSPO on the MathVerse benchmark.

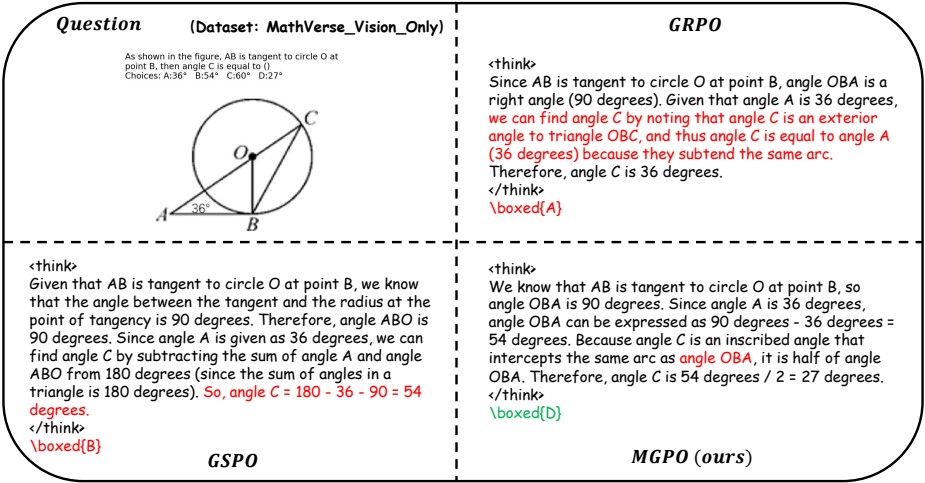

Figure 10: Case visualization of our method over vanilla GRPO and GSPO on the MathVerse benchmark.

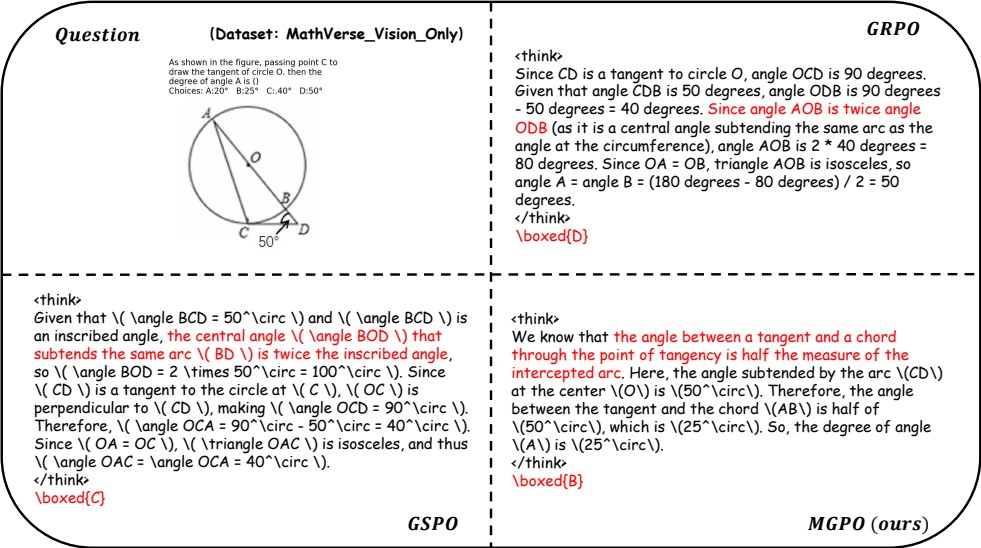

Figure 11: Case visualization of our method over vanilla GRPO and GSPO on the MathVerse benchmark.

