# OpenReview forum: "Multi-Grained Policy Optimization for Multimodal Reasoning: From An Uncertainty Perspective"
_ICLR.cc/2026/Conference — ICLR 2026 Conference Withdrawn Submission_

### Official Review · Reviewer_DLbS · 2025-10-16

**Soundness:** 2
**Presentation:** 3
**Contribution:** 3
**Rating:** 2
**Confidence:** 5

**Summary:**

This paper introduces MGPO, a reinforcement learning training method that incorporates a Multi-Grained importance sampling approach. It proposes Multi-Grained Integration and Uncertainty-aware Token Sampling. Experiments conducted under GRPO-like MLLM reinforcement learning training settings demonstrate that MGPO offers performance advantages.

**Strengths:**

The writing style of this paper is excellent and easy to follow, and the overall logic is very coherent.

**Weaknesses:**

1. The ideas in this paper are presented in a stream-of-consciousness manner and lack substantive analysis. What exactly are “training stability” and “token exploration”? The paper does not show results related to training stability compared with the baseline methods. Regarding token exploration, Figure 5’s entropy loss suggests that, compared to the GRPO baseline, this method actually reduces overall exploration.
2. As a general, modality-agnostic post-training reinforcement learning algorithm, this paper devotes substantial space and numerous experiments to hyperparameter tuning (e.g., Figures 2, 3, 4, and Table 4). I would prefer to see consistent improvements across diverse datasets and tasks (image QA, text QA, video QA). The authors don’t need to add too many extra experiments; if possible, they could start with text-only tasks first.
3. The paper lacks a summary of limitations and a plan for future work.

**Questions:**

1. I’m confused about the figure in the top-right of Figure 1. How is the Importance Sampling Weight computed there? GRPO uses clipping, so this weight should be within 0.8–1.2, but the results in the figure significantly exceed this range.
2. In Figure 2, when α equals 1, is it equivalent to GRPO? This seems to conflict with the results in Table 3.

---

### Official Review · Reviewer_Gtwo · 2025-10-20

**Soundness:** 3
**Presentation:** 3
**Contribution:** 3
**Rating:** 4
**Confidence:** 3

**Summary:**

This paper addresses limitations in reinforcement learning (RL) for enhancing reasoning in Multimodal Large Language Models (MLLMs). It builds on Group Relative Policy Optimization (GRPO), which uses token-level importance sampling weights but suffers from training instability and insufficient token exploration. Group Sequence Policy Optimization (GSPO) mitigates instability via sequence-level weights but reduces exploration. The authors propose Multi-Grained Policy Optimization (MGPO), which dynamically blends token-level and sequence-level weights using uncertainty estimation on log probabilities to balance stability and exploration.

**Strengths:**

The technical quality is robust, with a solid methodological foundation and comprehensive empirical validation. The algorithm is well-derived, starting from clear preliminaries on GRPO/GSPO and progressing to MGPO's objective function.

The paper excels in clarity, presenting complex RL concepts in an accessible, structured manner without overwhelming jargon. It begins with a motivating abstract and introduction that clearly outlines limitations of prior methods and poses a key question: "Is it possible to harmonize training stability and token exploration within a single system?"

**Weaknesses:**

the uncertainty-aware sampling draws heavily from prior uncertainty-driven RL methods like the "80/20 rule" (Wang et al., 2025a), which already emphasizes high-entropy tokens for LLM reasoning, and SEED-GRPO (Chen et al., 2025c), which modulates policy updates via semantic entropy across rollouts. Similarly, dynamic weighting echoes adaptive techniques in EMPO (Zhang et al., 2025b), where entropy minimization incentivizes reasoning. This combination is applied to multimodal settings, but the paper does not sufficiently differentiate how MGPO's uncertainty metric (-\log P_{i,t}) uniquely advances beyond these, potentially understating overlaps. To improve, the authors could add a dedicated section deriving a novel theoretical bound (e.g., on variance reduction in importance sampling under uncertainty guidance) or prove convergence properties specific to multi-grained weights, perhaps using tools from variational inference, to elevate the originality beyond synthesis.

All training uses only 2.1K samples from Geometry3K (Lu et al., 2021), a geometry-specific dataset, and evaluations are confined to math-related out-of-distribution benchmarks (MathVerse, MathVision, MathVista, WeMath). This raises concerns about domain specificity—e.g., no tests on diverse multimodal tasks like visual question answering or commonsense reasoning—potentially overfitting to numerical/geometric patterns.

**Questions:**

The paper motivates multi-grained weights intuitively but lacks formal analysis, e.g., on how the blending reduces variance or improves exploration bounds compared to GRPO/GSPO. Could you derive or sketch a theoretical bound (e.g., on KL divergence or policy gradient variance under uncertainty guidance) to support why this integration is superior to simple averaging?

---

### Official Review · Reviewer_YhHw · 2025-10-31

**Soundness:** 2
**Presentation:** 2
**Contribution:** 2
**Rating:** 4
**Confidence:** 3

**Summary:**

This paper proposes Multi-Grained Policy Optimization (MGPO), an RL algorithm that combines both sequence-level and token-level importance sampling to balance the stability of training and sampling exploration. Experiments show that MGPO outperforms Group Relative Policy Optimization (GRPO) and Sequence Policy Optimization (GSPO) in visual reasoning benchmarks.

**Strengths:**

1. The method is simple and easy to implement, which means that it has the potential to widely adopt in broad scenarios, from academia to industry.

2. This paper shows comprehensive empirical results on various benchmarks, compared with different baselines, which explicits demonstrate the performance of MGPO.

**Weaknesses:**

1. While the method is simple, the contribution is also limited. As a simple fusion of token-level and sequence-level importance sampling can not be called a novel method. It would be more like a technical implementation.

2. The motivation for selecting visual reasoning benchmarks is not clear. It seems like MGPO is a general optimization algorithm for both the text and visual domains. However, this paper discusses the visual reasoning as the background and does not involve an empirical study on text-based benchmarks.

3. The presentation of the paper could be improved. For instance, only a single figure (Figure 1) is provided to illustrate the MGPO method, and Algorithm 1 is shown solely as Python code rather than in a more formal or readable format.

**Questions:**

How sensitive is MGPO's performance to variations in hyperparameters like α and β, and does the dynamic α consistently outperform the fixed version across different datasets or model scales? Could the author provide some suggestions for the selection of $\alpha$ and $\beta$ in training?

---

### Official Review · Reviewer_5Vdy · 2025-11-01

**Soundness:** 2
**Presentation:** 2
**Contribution:** 2
**Rating:** 4
**Confidence:** 4

**Summary:**

This paper introduces Multi-Grained Policy Optimization (MGPO), an RL algorithm tailored for multimodal reasoning in vision-language models (VLMs). MGPO addresses two key limitations of existing methods like GRPO and GSPO—training instability and limited token exploration—by dynamically integrating token-level and sequence-level importance sampling weights using uncertainty estimation.

**Strengths:**

- MGPO introduces a novel integration mechanism that dynamically blends token-level and sequence-level importance sampling weights using uncertainty-based token selection
- The method is soundly formulated, with clear mathematical definitions and a concise algorithm
- The paper is well-structured, with intuitive explanations of GRPO/GSPO limitations and how MGPO addresses them.

**Weaknesses:**

- Limited Theoretical Justification
    - While the intuition behind blending token- and sequence-level weights is clear, no theoretical analysis is provided to justify why this blending leads to better optimization dynamics or convergence properties.
    - The choice of uncertainty metric (negative log probability) is heuristic. No ablation or comparison with other uncertainty measures (e.g., entropy, mutual information) is provided.
-  Narrow Scope of Evaluation
    - All experiments are conducted on geometry-style visual math tasks. It is unclear how well MGPO generalizes to other multimodal reasoning domains (e.g., chart understanding, visual question answering, or embodied AI).
    - The base model (Qwen2.5-VL) is fixed. It is unclear whether MGPO’s benefits transfer to other model architectures (e.g., LLaVA, InternVL). Moreover, how about employing MGPO into text-only LLMs？

**Questions:**

- Why does negative log probability serve as a good proxy for uncertainty? Have you tried other uncertainty measures (e.g., entropy, semantic entropy, ensemble disagreement)? How sensitive is MGPO to this choice?
- How does MGPO behave with other base models or architectures? Have you tried MGPO on LLaVA, InternVL, or Kimi-VL? Are the gains consistent?
- What are the failure modes of MGPO? Can you provide a categorical breakdown of failure cases (e.g., vision misperception, logical error, optimization collapse)? Are there systematic patterns?

---

### Note · Authors · 2025-11-14

I have read and agree with the venue's withdrawal policy on behalf of myself and my co-authors.